# Participation of Nursing Students in Evidence-Based Practice Projects: Results of Two Focus Groups

**DOI:** 10.3390/ijerph19116784

**Published:** 2022-06-01

**Authors:** Cristina Lavareda Baixinho, Óscar Ramos Ferreira, Marcelo Medeiros, Ellen Synthia Fernandes de Oliveira

**Affiliations:** 1Nursing School of Lisbon, Nursing Research, Innovation and Development Centre of Lisbon (CIDNUR), 1900-160 Lisbon, Portugal; oferreira@esel.pt; 2Center for Innovative Care and Health Technology (ciTechCare), Polytechnic of Leiria, 2411-901 Leiria, Portugal; 3Nursing School, Federal University of Goiás, Goiânia 74690-900, Brazil; marcelo@ufg.br; 4The Graduate Program in Collective Health, Federal University of Goiás, Goiânia 74690-900, Brazil; ellen@ufg.br

**Keywords:** evidence, knowledge translation, learning, nursing, students

## Abstract

The development of true evidence-based practice requires that practitioners have the knowledge and skills to research, analyze, and use evidence. These skills must be acquired in pre-graduate training. The objective of the present study was to analyze the contributions of students’ participation in knowledge translation projects to clinical practice for evidence-based learning. This was a qualitative, descriptive, and exploratory study that used focus groups. Scripted interviews were administered. The design of the study included five phases and took place in the partnering institutions of the Safety Transition Project, involving fifteen participants. The study was authorized by the Research Ethics Committee. The data were analyzed following the steps encoding the categories, storage and recovery, and (3) interpretation and using computer software (WebQDA^®^, Ludomédia, Aveiro, Portugal). Four categories were identified: learning evidence; communicating science; evidence-based practice; and developing skills. The successful implementation of evidence-based practice education resulted in students who understand its importance and use it competently. Further research should explore the skills developed by nurses involved in similar projects and their contribution to an EBP culture.

## 1. Introduction

Evidence-based practice (EBP) is recognized as key to improving the quality of health care [1], increasing patient safety and controlling costs and improving outcomes for people with healthcare and nursing needs [2,3].

However, a literature review showed that nurses are not implementing EBP at levels considered desirable and recommended by scientific and international organizations (International Council of Nurses, World Health Organization, Canadian Research Knowledge Network) that are responsible for issuing guidelines for health settings [1,4,5]. Some authors have indicated that this gap, between what is known and what is used in clinical contexts, is a global problem that is not restricted to a specific country or geographical area and that, therefore, the adoption of EBP has not yet become a standard inherent to healthcare provision around the world [1,2,6]. This results in a gap between research, practice, policy, and research outcomes [2,7,8], causing delays in the introduction of EBP in clinical settings.

The exponential increase in nursing research, with the resulting increase in its dissemination, has not been accompanied by the same rate of implementation in clinical practice settings [9]. Although the purposes of nursing research (to conduct research to generate new knowledge) and evidence-based nursing practice (to use the best evidence available to underpin nursing practice) seem quite different, they are in fact part of the same process that allows the new knowledge generated by the scientific method to be introduced into the clinic, producing cost-effective health care based on the most up-to-date knowledge [10].

The use of knowledge in health is not a recent concern among researchers because of the delay in the introduction of the results of research in some clinic areas, justified by some authors because of methodological and ethical issues, scientific rigor, project execution capacity, research funding challenges, lack of an EBP culture among the organizations [2,7,8,9], and lack of a scientific culture of collaborative work to develop products that promote the introduction of results in clinical settings [2,9]. Other factors are a preference for unidirectional models to introduce research results to issues related to education and the lack of opportunities given to students in undergraduate programs relative to this type of learning [2].

Internationally, EBP is recognized as a crucial element for the education of all health professionals [11]. However, achieving skills in this area is a complex activity that is reflected in the disparities between “what the evidence says” and “what is done in clinical practice” [9,11]. Although there are recommendations for teaching strategies to improve knowledge and skills in this area, the development and implementation of vocational education programs remain a significant and immediate challenge [11] because insufficient attention has been given to its use in nursing education [3].

Recent studies have reinforced that nurses are not well prepared to apply EBP [3] and that nursing students do not recognize the importance of evidence and do not have enough knowledge and skills to use it [5]. As stated by Larsen et al. (2019), the Sicily statement recommends a five-step model in order to meet the minimum requirements for teaching and conducting EBP: (I) asking a clinical question; (II) collecting the most relevant evidence available; (III) critically appraising the evidence; (IV) integrating evidence into clinical knowledge, patient preferences, and values to make practical decisions; and (V) evaluating changes or outcomes [4].

Considering that this is an appropriate recommendation that guides health professionals in clinical practice [5], we agree that it is imperative to instill passion and enthusiasm among students regarding research and its daily relevance in ensuring quality of care [12] and cost-effective health outcomes [13].

The development of EBP competencies implies the acquisition and consolidation of a minimum set of attributes, knowledge, skills, and attitudes [14]. Clinical teachings are a key time to improve students’ knowledge, attitudes, and skills in EBP [15,16]. Learning about research and the use of scientific knowledge is more significant when integrated into this type of teaching [2].

In Portugal, we have a knowledge translation (KT) project for the clinic that involves nursing students. From the dialogue between our experience and the literature review, the research question of this study emerged: What is the perception that nursing students have about the contribution of their participation in KT projects for the learning and development of EBP skills?

In light of the above, the objective of this study was to analyze the contributions of student participation in projects of knowledge translation to clinical settings for evidence-based learning.

## 2. Materials and Methods

### 2.1. Study Design

This was a qualitative, descriptive, and exploratory study that used focus groups (FGs) as a method to answer the following research question: “What do nursing students learn about EBP when participating in knowledge translation projects during their professional integration internship?”. The study design consisted of five phases [17]: planning, preparation, moderation, data analysis, and dissemination of results.

This methodological option is justified because: (1) the literature review demonstrated that there are few studies about the participation of undergraduate nursing students in research projects; (2) the uniqueness of the experience that was provided to students justified exploring its contribution to EBP learning; and (3) focus group discussions are the preferred method when descriptions, meanings, opinions, and reflections are the sources of knowledge about a phenomenon or a specific situation or issue [18]. Focus groups promote interaction among participants and can result in even deeper and more consistent knowledge, based on the sharing of experiences from different points of view and confrontation of opinions [17,18].

To achieve the objective of the study and increase its depth, two FGs were carried out with two different groups of participants who shared the same experience of undergoing the last internship of the undergraduate nursing program.

### 2.2. Settings and Participants

The study took place in the partner institutions of the Safety Transition (ST) Project, which is a knowledge translation program that involves a partnership between a hospital, a nursing school, and a primary health care service in the region of Lisbon and the Tagus Valley in Portugal.

The ST Project is based on a simultaneous process of problem solving, training, research, and action, in which the primary purpose is the use of knowledge translation to solve the problems of different hospital inpatient services, increasing the knowledge of and empowering patients and their families in the process of transitioning from the hospital to the community. The project establishes communication circuits that promote continuity of care, reducing readmissions after hospital discharge and promoting rehabilitation and insertion in the community [19,20].

Over the course of this project, students can be integrated into hospital inpatient services (emergency room, intensive care unit, medicine, surgery, and orthopedics) or into primary health care services, where the last semester-long undergraduate nursing internship takes place.

The choice to involve students is based on the consensus that achieving a high level of EBP learning requires investing in quality clinical practice environments and improving the clinical experiences of students, not least because the clinical settings and situations of patients are becoming increasingly complex, requiring a higher level of scientific skills [21].

The participants were selected from among the students involved in the ST Project in the second semester of the 2018/2019 academic year. Similar to other qualitative methods, FGs rely on intentional samples. Therefore, participants are selected based on the possibility of generating the most productive discussions [17]. To be included, students had to have completed their last internship of the program in one of the project’s partner institutions, been involved in one of the ongoing projects in different services, have obtained a passing grade in the curricular unit in previous years, and have consented the participation. The sample was homogeneous and allowed the focus to be on discussion of the phenomenon under study, because they all shared the characteristic relevant to the topic under discussion—being part of a knowledge translation project in clinical practice—which reinforces the observations of Krueger and Casey about the selection of the constituents of a discussion group [18].

The number of participants was established a priori according to the ideal number of participants defined by the same authors, between five and ten participants [17,18]. At the beginning of the internship (clinical teaching), all the students involved in the project were informed, during a meeting with the principal research, that at the end they would have the opportunity of participating in an FG, with their voluntary and informed consent. Two weeks before the end of the internship, potential participants were given information regarding the date of the FGs and were asked to confirm whether they were still interested. Fifteen students confirmed their interest, and they were randomly distributed between the two FGs.

### 2.3. Data Collection

The literature review was crucial for structuring the interview script, which was organized around the following question: How has being involved in the Safe Transition Project contributed to your learning? Based on this initial question, the researchers were able to taper the discussion down to more specific questions [18]. Both the central and secondary questions helped guide the discussion, but they were not intended to restrict it; as the FGs progressed, more specific issues arose depending on the groups’ contributions. This approach is committed to group discussion, and as such, the first part allows the researcher access to the perspectives of the participants themselves, while the second part provides answers to the researcher’s specific interests [17,22,23].

The data collection sessions were video-recorded and took place at the school of nursing, in a room organized for this purpose, creating a comfortable environment that fostered participation, prevented interruptions, and ensured confidentiality [17,18].

In the e-mail sent to students and at the beginning of the focus group, participants were informed about the objectives and rules of participation, including estimated duration (75 min), avoiding early dropouts at the start of the group discussions [17]. At the beginning of the sessions, the moderator and co-moderator identified themselves.

The role of the moderator is key in this phase [18], not only to support the group in exploring the topic, but also because of the possibility that new insights may arise [17]. The guiding questions and the work of the moderator were previously reviewed by the team to ensure the necessary moderation skills, group dynamics, and control of possible critical elements to ensure a successful FG [17].

The choice of a co-moderator was in line with the recommendations of Krueger and Casey and was designed to increase the rigor of the process. The moderate mission was to develop discussion and keep it going. The co-moderate mission was to help the moderator manage the recording equipment, be aware of the conditions and logistics of the physical setting, respond to unexpected interruptions, and take notes on the group discussion [18].

### 2.4. Data Analysis

The recording was watched twice before being transcribed by one of the researchers present in the FGs so that they could “visualize” what had occurred in the group. The findings were analyzed according to the steps defined by Bloor, Frankland, Thomas, and Robson [24]: (1) encoding of the categories, reflecting the themes present in the transcripts and new ones that emerged from the group discussions; (2) storage and recovery, used to extract all the statements from the text assigned to the same category for comparison; and (3) interpretation, based on data analysis. Representativeness, comprehensiveness, homogeneity, and relevance were ensured when the categories were defined.

The data were analyzed using computer software (WebQDA^®^, Ludomédia, Aveiro, Portugal), which helped organize and analyze the findings and increased the rigor of the process. Coding was carried out by the researcher who transcribed the recordings of the FGs, and then it was validated by the research team.

A code was assigned to each participant (P1, 2, 3…) in both FGs (FG1, FG2).

The research phases were rigorously conducted and validated by the entire team so that the results accurately represented the participants’ experiences. After coding the findings, they were returned to the participants for their validation, ensuring the credibility of the study. Confirmability was guaranteed by the communication that took place during the coding process, by the literature, among the team, and by an expert who evaluated the codes that emerged from the findings. Transferability was demonstrated by the depth of the analysis, the methodological description, and the presentation of the results, which increase the likelihood of the findings being significant in other similar contexts.

### 2.5. Ethical Considerations

This study was authorized by the Ethics Committee of Hospital Vila Franca de Xira (Protocol number: 09/2019 HVFX). The institutions involved have a formalized partnership protocol that authorizes research. All ethical and formal principles were respected, from authorization for the students to integrate the project as an extracurricular activity, to the ethical issues inherent in the development of research. All participants were assured of the anonymity and confidentiality of the data.

Since this was an extracurricular activity that took place simultaneously with a clinical internship resulting in a final grade, the FGs were only carried out after the participants had received their grades for that curricular unit to minimize the effect of “socially desirable” responses.

## 3. Results

Eight students participated in FG 1 and seven students in FG 2. They were mostly women (12), and most had carried out their internship in the hospital setting (14) (Table 1).

In response to the question about EBP learning based on their participation in the ST Project as part of their professional integration internship, 192 registration units (UR) emerged in the participants’ discourse, organized into four categories: learning evidence (n = 94); communicating science (n = 22); evidence-based practice (n = 55); and developing skills (N = 21) (Table 2).

### 3.1. Learning Evidence

Evidence is a complex concept associated with the generation (production), synthesis, transfer, and implementation of knowledge for health care. Learning evidence associates the need to acquire knowledge about the different phases but also to develop skills to apply it in the clinic.

In relation to the category “learning evidence,” the participants indicated that being involved in the project had given them the opportunity to look up research articles, read them, and evaluate the quality of publications, i.e., level of evidence. This opportunity to read science was mentioned in statements such as:


*I think it’s something that was complemented, better than reading 100,000 articles and being overwhelmed was reading and seeing how they could be applied in practice, i.e., verifying the applicability of what we looked up and what was researched, so it no longer was just an article but something with a result, a solution of interest.*
(FG2, P.1)


*In conversations with colleagues who were from other institutions, I realized that they did not use the databases with the same rigor as we did. Research and quality analysis of the articles was important for us to appraise their quality and verify what they actually presented in terms of evidence. Furthermore, I stopped accepting everything as fact or correct, when they tell me “it’s done this way” I always ask why and the basis behind it.*
(FG1, P.1)


*We end up wanting to continue, we do the research, apply the findings, see the results and want to keep going.*
(FG1, P.4)

The opportunity to search databases and read scientific articles was recognized as important for the promotion of a scientific and/or technical culture. This type of reading stimulates the cognitive and sensory processes of perception, curiosity, and motivation to analyze the contribution of results to the development of research projects, contributing to the meaning of research for problem-solving.

Learning how to read and evaluate scientific articles is accompanied by the possibility of synthesizing evidence by developing systematic or integrative literature reviews:


*Literature reviews based on the scientific method were a great lesson, not only in how research is conducted, but what I struggled with most, which was evaluating the quality of the research. Before this internship we searched for articles and if they were in the school database, we assumed they presented evidence. Here we learned that it is much more than that, we had to separate the wheat from the chaff, I think that’s the proverb, we had to verify the quality of the studies and the levels of evidence (…) reviewing the literature allowed us to theoretically substantiate the evidence that supports the project and not just that, (…) it helped us improve our follow-up consultations, including the instruments used to evaluate patients.*
(FG2, P. 3)

The enormous quantity of existing research justifies the need to synthesize evidence, and professionals can play an important role in this process, or in its use, as the end consumers of science. This synthesis is a challenge and future professionals should develop skills in this area. The students indicated that they developed attitudes that favored this type of learning and research and synthesis skills.

In relation to theoretical-practical integration, the FGs revealed the students’ perceptions that the experience had contributed to the integration of theoretical and practical knowledge:


*There were things that I saw in practice but that I better understood in theory, and there were things in theory that I understood better once observing them in practice.*
(FG2, P. 1)


*I actually mentioned this to my colleagues, I was able to observe the practice and I was able to understand the applicability of that, of our work. In that respect, the internship was very good, and for me, it was very gratifying to see that we were doing things that we knew were being applied.*
(FG2, P.6)


*For me, it helped narrow the gap between theory and practice. It might seem strange to say this, but often we complain that the content in school is too theoretical and little applied in practice, there are huge gaps, but this experience allowed us to grasp the theory and apply it to practice.*
(FG1, P.8)

The relationship between theory and practice is a recurring issue in nursing education, often based on a duality, which, as is clear in the participants’ discourse, is based on a separation between ways of doing and ways of thinking. The gap between theory and practice was reduced because the project allowed students to introduce knowledge into the clinical setting.

This theoretical-practical integration contributes to evidence-based clinical decision-making. The future nurses emphasized the following positive points:


*This experience has allowed me to see what EBP is very concretely, EBP requires research, method, palpable results, it’s not a philosophy of opinions, it involves concrete results obtained through rigorous research processes, which is very important, and this is how it has benefitted me, through the effective understanding of EBP, it is the result of things that are well done and that have been reached in a right and correct way and… that’s it! In my opinion, we talk a lot about evidence without knowing very well what it is.’*
(FG2, P.6)


*It allowed us to develop what we were working on further, and throughout this work, there was always a bridge between what was reality and what maybe was science, that we could be doing more, that there was positive evidence about how to do it and this contributed to our reflection about maybe what little things could make a difference in our practice.*
(FG2, P.4)


*In school, we learn the ideal, what has to be done, but when we arrive in clinical practice, there are so many challenges, lack of time, or resources, both human and material, and we end up seeing that there is a lot that is not mentioned in school, in terms of the difficulties we will face as professionals and how to resolve them, school doesn’t talk much about this, everything is very ideal.*
(FG2, P.7)

Negotiating decisions as a team, clarifying the intention behind decisions, anticipating outcomes, reflecting about actions, and problem-solving capacity all contribute to EBP learning and certainty.

Learning about evidence is also achieved through opportunities given to students to transfer their knowledge into the clinical setting, which was expressed by the participants as follows: 


*What I mean is that we have an idea of the evidence and that it is used in clinical practice, but in reality it’s still hard to support everything with evidence, even when guidelines come out determining that things should be done a certain way, practices are hard to change, it is ongoing work, almost baby steps.*
(FG2, P.5)


*We studied this in theory and with the work developed we were able to transfer it into practice. We even saw how some providers worked in other countries in order to adapt it to our reality, based on what is written about whether the outcomes are suitable to people’s needs.*
(FG1, P.3)


*I understand that people would like to do this, but these things take time and this project showed me that it takes not only time, but also a methodology determining what has to be done, as my professor used to say, you can’t introduce evidence with a magic wand, you need a structured, objective, simple plan and to think about the benefits for patients.*
(FG1, P.1)

Knowledge translation is seen by the students as a difficult process that they had not been aware of until the internship. Their involvement in the project increased their recognition of the value of scientific evidence, and it is this recognition that makes knowledge translation add value to health care.

### 3.2. Communicating Science

Science communication implies knowledge of research methods and techniques, the ability to methodologically evaluate studies and synthesize the results. It requires scientific communication skills, written and oral, and the ability to adapt the language to the target audience, which in the EBP is, most of the time, the clients of health care.

Regarding the category of communicating science, the students said:


*In my opinion, based not only on my work, but that of my classmates, creating the posters was a way of synthesizing information, but it was also challenging because there was a lot of information and we had to be able to summarize it in the most pertinent way in order to put it on a poster.*
(FG 1, P.6)


*It was a very important time, because it forced me, not only to learn how to use scientific language, but also data presentation strategies, and we prepared ourselves ahead of time to answer hypothetical questions that could arise, because often we have data analyses with their conditioning factors and we need to know how to answer questions that might be made not only in terms of data communication but also about the conditioning factors found and how we resolved them.*
(FG1, P.7)

The students reinforced the importance of their learning from the experience of creating posters, collaborating in drafting scientific articles, and discussing them with other people. Science communication is a bilateral process that involves communication skills, the sharing of ideas, the synthesis of information, and the use of scientific language.

### 3.3. Evidence-Based Practice

EBP implies the use of the best available evidence to improve health care, which implies to change the clinical practice, improve patient outcomes, improve health indicators, change health policies, and create an evidence ecosystem.

The participants also spoke about the opportunities for improving care practices that arise when translating knowledge into clinical practice. Regarding this aspect of EBP, the students explained that:


*We did other things than care delivery, I thought it was very interesting to observe the nurses involved in projects, improving care, concerned with the outcomes of their interventions, and how to use evidence.*
(FG1, P.1)


*I realized that theoretical work can be implemented in practice and it’s satisfying seeing that the results we obtained with the integrative review could actually be used and it wasn’t just an assignment and that’s it. In this sense it was very gratifying because we knew our research would produce results, would change things in clinical practice, and we had the possibility of participating in this change, because we had begun to implement the project into the service with what we had and began to see changes. That was very encouraging, because we also began to want to know whether the nurses were adhering to this change, what the results are, and what has to be reformulated.*
(FG1, P.5)

The applicability of the projects in the clinical setting fostered the improvement of the quality of care provided to patients, in the short or medium term. By going through this experience in a practical setting, the students increased their awareness of the benefits of EBP for users and services.

### 3.4. Developing Skills

Third generation soft skills or competencies are important for EBP, because they imply interpersonal relationship skills, leadership, teamwork, active listening, and time management, among others that allow the ability to adapt and change that are necessary to work in evidence ecosystem.

Finally, the participants demonstrated that they recognized the importance of developing cross-sectional skills relative to professional autonomy, time and conflict management, and leadership, expressed as follows:


*I felt I was the leader in some respects, I had to plan meetings, prepare content, negotiate with staff nurses what would be addressed in these meetings. It also meant additional research, exploring what was happening in other institutions, I felt competent in so many activities, that I was working side-by-side with professionals. I realized above all that there is work beyond what we usually do with patients in internships, it was very gratifying!*
(FG2, P.7)


*It allowed for good time management, which was related to everything because there are so many of us, sometimes it’s hard to organize everybody’s schedule, and I felt that if there is time for us to truly focus on what this is and take advantage of opportunities, this will produce more learning outcomes.*
(FG1, P.7)

Cross-sectional skills develop in parallel when students can carry out the activities included in the projects, increasing the possibilities for students’ active participation in the projects and in clinical decision-making.

## 4. Discussion

The results of the present study reinforced the results of other research that has found that the implementation of successful EBP education is necessary for students, not only to understand the importance of EBP, but also to be competent in its use, basing their decisions on research results. In essence, the students’ experience contributed to developing professionals who value EBP and have the knowledge and skills to implement this practice, with the ultimate goal of enhancing the delivery of health care for improved patient outcomes [11].

Creating an effective learning experience enables the understanding of evidence, scientific communication, evidence-based practice, and the development of skills. These findings corroborate the notion that active learning experiences that are well structured and that work closely with clinical contexts help construct nursing knowledge [25,26], increase students’ EBP knowledge and skills [3], expand possibilities for future use, strengthen critical thinking, and improve the capacity for healthcare provision [5].

In view of the above, some experts emphasize that joint and active work between academic and clinical educators across healthcare professions is necessary to foster a “real world” pragmatic approach to the integration of EBP education [11].

A systematic literature review identified two key methods for EBP education among nursing undergraduate students: participation in research courses and workshops, and their collaboration with clinical practice, suggesting the incorporation of the five steps of the Sicily statement in these activities [4]. Despite this recommendation, Horntvedt et al. affirmed that collaboration with clinical practice to improve knowledge of EBP is not always a reality [3], and according to Lehane et al., further attention is needed to strategies that not only focus on issues such as curriculum structure, content, and program delivery, but also support educators, educational institutions, health services, and clinicians in developing the capacity and competence to meet the challenge of providing EBP education [11].

Given the above, organizations and nursing leaders can benefit from the development of a comprehensive strategy to promote the involvement of nursing staff in the EBP process by providing continuous education and mentoring programs [1]. Further studies should explore the competencies developed by health professionals involved in projects and whether this contributes to an evidence culture in healthcare institutions.

The content analysis of the discourse of the participants in the two FGs showed that a close working relationship between the educational institution and the clinical settings, the possibility of ‘hands-on’ participation in the projects, and the knowledge translation that emerged in these contexts added value to their research, readings, evidence synthesis, and transfer of knowledge. It also showed the impacts that EBP can have on enhancing care, which tends to confirm that there are important skills and competencies to be learned for EBP, such as evidence-based decision-making, knowing how to ask questions, knowing how to translate knowledge into clinical practice, evaluating research, interpreting results, and communicating science, as has been observed in other studies [2,11,26].

It is important that students recognize the applicability of the activities they are involved in and that they teach evidence-based decision-making and knowledge translation into clinical practice. Clearly, to achieve significant results, evidence that stems from research (effective innovation) must be accompanied by effective implementation and an enabling context [10]. Future studies should deepen the contribution of the project to the introduction of EBP and its future impact on the attitudes of these professionals regarding the use of EBP.

It is worth noting that, in the evaluation of research initiation programs developed by Slattery et al. [27] and Cardoso et al. [2], the findings indicated that early exposure of nursing students to different research activities encouraged recent graduates to incorporate evidence into their future clinical practice and to be more proactive in seeking out graduate training. Other studies on the subject have observed the advantages of active participation in the different phases of research, but have not clearly shown the impacts of these experiences after their completion.

We agree that EBP teaching should be a priority in nursing programs and should be translated into academic and clinical curricula [5,11].

### Limitations and Strengths

This study has limitations associated with the nature of the study and the method and data collection technique. The intentional choice of participants and the concrete experience of having participated in the SF project limits the results to the context. The interaction between participants of the FG may have influenced the response to what is socially accepted.

Despite the aforementioned limitations, it is reinforced that the qualitative nature of the study allowed us to understand the students’ perspective on the potential of the SF project for the development of skills necessary for EBP. The use of the qualitative analysis software of the findings allowed the organization of the findings and contributes to increase the rigor in the process of analysis, coding, and selection of illustrative excerpts of emerging categories.

## 5. Conclusions

The two focus groups allowed us to analyze the contributions of the nursing students’ participation in EBP learning through knowledge translation projects in clinical practice. The discourse analysis of the fifteen participants allowed the definition of four categories: learning evidence (n = 94); communicating science (N = 22); evidence-based practice (n = 55); and developing skills (n = 21).

The results of this study contribute to the public debate on the importance of EBP in nursing curricula and the need to coordinate the actions of academia and clinical contexts in order to promote the learning and development of skills that increase the scientific literacy of students and contribute to the development of an EBP culture in healthcare institutions.

We corroborate the need to introduce content related to EBP in the curricula, using active methodologies that enable the development of skills for the research, synthesis, use, and implementation of evidence in clinical contexts of high complexity.

## Figures and Tables

**Table 1 ijerph-19-06784-t001:** Demographic data of the participants.

	Focus Group 1	Focus Group 2
Male	1	2
Female	7	5
Age	21.7 (±1.9) year	22.3 (±2.4) year
Service		
Medicine 1	1	1
Medicine 2	1	0
Intensive Care	2	2
Orthopedics	1	2
Emergency Room	2	2
Primary Care	1	0

**Table 2 ijerph-19-06784-t002:** Corpus of content analysis.

Category	Subcategory	UR
Learning Evidence	Reading Science	27
	Synthesis of Evidence	16
	Theoretical-Practical Integration	15
	Evidence-Based Decision Making	14
	Knowledge Transfer	22
	**Subtotal**	**94**
Communicating Science	Communicating Evidence	16
	Scientific Writing	6
	**Subtotal**	**22**
Evidence-Based Practice	Opportunities for Improvement	34
	Promoting Better Care	21
	**Subtotal**	**55**
Developing Skills	Autonomy	4
	Time Management	10
	Conflict Management	5
	Leadership	2
	**Subtotal**	**21**
	**Total**	**192**

## Data Availability

The data used during this study are available from the corresponding author, under request by e-mail.

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
