# Peer review of "Participation of Nursing Students in Evidence-Based Practice Projects: Results of Two Focus Groups"

_ijerph, 2022, doi:10.3390/ijerph19116784_

Round 1

Reviewer 1 Report

Participation of nursing students in knowledge translation 2 projects in clinical practice: results of two focus groups

1.Title: Long title exceeding the amount of up to 15 words. It is suggested to clearly explain the object of the study. It is suggested to remove the term knowledge translation, since the focus throughout the study is on evidence-based practices, as well as readjustment of the title consistent with the object of study.

2. Abstract: It is presented with an incongruous structuring in a phrasal topic when starting with suggestive elements of discussions or final considerations. The other segments are adequate, with a clear description of the objective, method, results and conclusions.

3. Introduction: Well-designed introduction, with conceptual emphasis and consistency with the object of study, presents connective sequencing between paragraphs and thematic clarity. It contextualizes the object of study, which is evidence-based practice, portraying the relevance of the theme, international perspective, and describes the justification in a cohesive way. However, it is suggested to highlight the research question in advance of the description of the objective and it is considered that in this way it constitutes an essential element for harmonic linearity between what is problematized and what is sought.

4. Method

The authors approach the methodology in a clear, objective and sequential way in explaining each step outlined by the chosen method, as well as justifying the choice of this method. They cite the methodological approach, define the research scenarios and participants, as well as inclusion criteria. Exclusion criteria do not exist. It is suggested that ethical aspects anticipate the description of the participants. Data analysis was duly described and aligned with the content analysis proposal, which involves the pre-analysis, material exploration, results processing, inference and interpretation steps. The authors used webQDA ® software (webQDA – Qualitative Data Analysis Software) is a web-based qualitative data analysis 5. Results The results show consistency in the categories listed, in accordance with the content analysis proposal. Table 1 indicates the participants' characterizations and table 2 describes the categorization of findings, including subcategories. It is considered that the description of the results is similar to clippings of the interviewees' speeches, making the description by category important. It is suggested to review the presentation of this result (cutting of speeches) emphasizing in this aspect to articulate the authors' perspective from the results found. 6. Discussion It approaches a synthesis of the results and brings aspects of the scientific literature in comparison with this one, pointing out similar aspects of outcome. Are there any discrepancies? Were there limitations and problematization? 7.Tables No considerations and/or suggestions 8. Conclusions/Final Considerations The conclusion in its first paragraph brings aspects related to results. Giving only from its second paragraph a conclusive reflection of the study, describing contributions. 9. References Features up-to-date and international references. Seem The present study has scientific and academic relevance, which highlights the importance of its contributions to nursing education. The method used is coherently aligned with the object of study. It is suggested structuring in the descriptive condition in its stages, in order to allow clarity and methodological alignment, according to the chosen approach, namely: - Situating the research question in advance of the objective description; - Anticipate the ethical aspects of the description of the participants; - Review the presentation of the result (cutting of speeches) emphasizing, in this aspect, to articulate the authors' perspective from the results found; - In conclusion redirect your first paragraph to results. We approve with reservations and we are available for any questions.

software intended for all researchers and professionals who carry out qualitative research. webQDA allows the analysis of text, image, video, audio sources, tables, PDF files, Youtube videos, etc. collaboratively, synchronously or asynchronously. Another relevant point was the validation by the participants of the findings, giving credibility to the study.

Author Response

Dear reviewer:

Thank you for taking the time to review our article and for the contributions that made it possible for us to improve.

We've introduced all the suggested changes and shaded it in yellow.

The only exception was for the recommendation of Anticipate the ethical aspects of the description of the participants because we consulted some articles published in the journal and the norms for publication and the recommendation is that they are in the section related to ethical aspects.

Reviewer 2 Report

Thank you for the opportunity to review this interesting work. Here are my comments for improvements.

Abstract:

1. Abbreviate evidence-based practice in its first appearance in the abstract then use EBP throughout

2. Why cross-sectional? This was not explained in the methods section.

3. What protocol? Is there a specific name as it was described well in the methods section.

Introduction:

1. Lines 35-37 - Is this a literature review that you published? Why you claimed that it's yours?

2. Lane 38 - What is referred by "this"?

3. Lane 36 - What international organizations?

4. Lane 45 - What purposes?

5. Lane 47 - Why "seem quite different" and what is its connection to the remaining phrase of the sentence in Lane 48?

6. Lane 50 - Why it is a concern?

7. Lines 50-55 - Needs significant revision for clarity.

8. Paragraph (lines 50-58) and paragraph (lines 59-65) - What is the connection of the two paragraphs?

9. What is the connection of lines 66-67 and 68-73?

10. Lane 67 - What approach?

11. Lane 74 - Is this same approach in lane 67?

12. Lane 76 - What research are you referring?

13. Lane 79 - "Internships" - The context of internship in the research settings must be presented here for better readership.

14. There is a misuse of terms in this section like EBP, EBP learning, EBP adoptation, etc... Why not start presenting EBP in general then down to EBP learning of nursing students.

15. There needs a significant revision in this section to justify the need to conduct the study.

Materials and Methods:

1. There needs a detailed description of the study protocol.

2. What are the benefits of ST project to nursing students?

3. What sampling technique did you use?

4. Lines 133-134 - Expressed or consented?

5. Lane 144 - Details on how the information was given.

6. After abbreviating Safety Transition Project, use ST Project throughout.

7. Lines 157-159 - Consider as two sentences.

8. Describe separately moderator and facilitator with their roles. Did you consider their qualifications?

9. Ethical considerations must be related to how you observed autonomy, beneficence, non-maleficence, and justice.

10. A separate section for the rigor of the study after Ethical Considerations section.

Results:

1. Lane 215 - What are registration units for clarify to readers?

2. Why subcategories were not presented?

3. Why use categories and subcategories, and not major themes and sub-themes?

4. Results need to be presented in an organized manner by category with its subcategories or major theme with its sub-themes. There needs a significant organization in this section.

Discussion:

1. Limitations and strengths of your study must be presented at the part before Conclusions section.

Conclusions:

1. Lane 450 - What was your concluding analysis?

2. Additional implications are need related to internship and ST Project to both nursing academe and nursing practice and policy.

Author Response

Dear reviewer:

Thank you for taking the time to review our article and for the contributions that made it possible for us to improve.

We've introduced all the suggested changes and shaded it in yellow.

We use the designation categories and subcategories according to theoretical referential used for the content analysis.

Reviewer 3 Report

The study is relevant and well written.  I also believe the study addresses a gap in the research.  The one area that I recommend minor revisions is the results section.  I believe the four categories -learning evidence, communicating science, evidence-based practice, and developing skills should have a clear definition based on the authors' interpretations.  Even though there is a table, a definition for each before the quotes would help give a clear definition of the terms based on the authors' ideas.

Author Response

Dear reviewer:

Thank you for taking the time to review our article and for the contributions that made it possible for us to improve.

We've introduced all the suggested changes and shaded it in yellow.
